# Racial Discrimination to Bullying Behavior among White and Black Adolescents in the USA: From Parents’ Perspectives

**DOI:** 10.3390/ijerph19127084

**Published:** 2022-06-09

**Authors:** Jun Sung Hong, Dong Ha Kim, Robert Thornberg, Sebastian Wachs, Michelle F. Wright

**Affiliations:** 1School of Social Work, Wayne State University, Detroit, MI 48202, USA; 2Department of Social Welfare, Chungwoon University, Hongseong 32244, Korea; dhkim@chungwoon.ac.kr; 3Department of Behavioural Sciences and Learning (IBL), Linköping University, 581 83 Linköping, Sweden; robert.thornberg@liu.se; 4Department of Educational Studies, University of Potsdam, 14476 Potsdam, Germany; wachs@uni-potsdam.de; 5National Anti-Bullying Research and Resource Center, Dublin City University, D09 AW21 Dublin, Ireland; 6Department of Psychology, DePaul University, Chicago, IL 60604, USA; mwrigh20@depaul.edu

**Keywords:** academic disengagement, bullying, depression, racial discrimination, race

## Abstract

The present study proposes and tests pathways by which racial discrimination might be positively related to bullying victimization among Black and White adolescents. Data were derived from the 2016 National Survey of Children’s Health, a national survey that provides data on children’s physical and mental health and their families. Data were collected from households with one or more children between June 2016 to February 2017. A letter was sent to randomly selected households, who were invited to participate in the survey. The caregivers consisted of 66.9% females and 33.1% males for the White sample, whose mean age was 47.51 (*SD* = 7.26), and 76.8% females and 23.2% males for the Black sample, whose mean age was 47.61 (*SD* = 9.71). In terms of the adolescents, 49.0% were females among the White sample, whose mean age was 14.73 (*SD* = 1.69). For Black adolescents, 47.9% were females and the mean age was 14.67(*SD* = 1.66). Measures for the study included bullying perpetration, racial discrimination, academic disengagement, and socio-demographic variables of the parent and child. Analyses included descriptive statistics, bivariate correlations, and structural path analyses. For adolescents in both racial groups, racial discrimination appears to be positively associated with depression, which was positively associated with bullying perpetration. For White adolescents, racial discrimination was positively associated with academic disengagement, which was also positively associated with bullying perpetration. For Black adolescents, although racial discrimination was not significantly associated with academic disengagement, academic disengagement was positively associated with bullying perpetration.

## 1. Introduction

Both racial discrimination (being treated unfairly or differently due to racial/ethnic status) [1] and bullying (unwanted, aggressive behavior among children and adolescents that is repeated and is characterized by power imbalance) [2] are serious concerns. An accumulated body of research found that between 10% to 33% of adolescents reported being a victim and 5% to 13% admitted to being a perpetrator of bullying in school [3]. Additionally, 23% of Black students and 23% of White students in U.S. public schools reported being bullied at school, according to the National Center for Educational Statistics [4]. Moreover, according to one study [5], which relied on data from the Pew Research Center, between 50% to 75% of Black people, Hispanic/Latino people, and Asian people reported feeling discriminated against by their peers.

Given the relevance of racial discrimination and bullying among racial and ethnic minority youth, studies have explored how racial discrimination might be linked to aggressive behavior. For instance, Wright and Wachs [6] examined the association between racial discrimination by peers and aggression among a sample of Latinx adolescents and concluded that racial discrimination is a chronic stressor that relates to aggressive behaviors. Bogart et al. [7] also reported that Latinx youth reported higher levels of delinquency, aggression, and retaliatory behaviors when they experienced higher levels of racial discrimination than their White counterparts. Another study conducted among indigenous adolescents from the Northern Midwest and Canada showed that early experiences of discrimination and anger each had indirect effects on aggressive behavior three years later [8].

As indicated in some studies, Black youth show a higher prevalence of bullying behaviors relative to their peers of other racial and ethnic groups [9,10,11,12]. Other study findings, on the other hand, suggest that racial-majority students are at a significantly higher risk of bullying behavior than racial-minority students [13]. A higher prevalence of bullying behavior among Black students in school might be due to biased perceptions of classmates and teachers in the peer nominations and teacher reports [14]. However, it is also possible that experiences in racial discrimination might reinforce bullying behaviors among Black adolescents, who are often victims of racial discrimination [5] and bullying by their peers [15]. Similarly, among racial-majority youth, being discriminated against may induce aggressive behaviors, as studies show a high prevalence of race-based victimization among this population. According to the Federal Bureau of Investigation [16], 20.1% of victims of hate crimes were due to anti-White bias, the second-highest rate of racial discrimination. Adolescents who are repeatedly mistreated by their peers due to racial/ethnic identity or skin color may act out aggressively to protect themselves from further victimization, especially when there is a lack of viable solutions [17].

### 1.1. Depression and Academic Disengagement as Mediators

Although a positive association between racial discrimination and bullying perpetration has been documented in the research literature, it is important to also understand that victims of racial discrimination are unlikely to engage in bullying immediately. Instead, they likely experience psychosocial problems, which may trigger problematic behaviors including bullying and aggressive behaviors. An example of this is depressive symptoms, which are frequently reported by victims of racial discrimination [18,19]. According to Stein et al.’s [18] findings, culturally-based stressors such as racial discrimination play a significant role in the development of depressive symptoms in Latino adolescents. Umaña-Taylor et al.’s [19] findings, also from a sample of Latino adolescents, indicated that greater online racial discrimination predicted depressive symptoms. Theoretically, experiences in racial discrimination may contribute to the development of a negative self-evaluation, which leads to depressive symptoms among adolescents [20] who are in the process of developing their identity and turning to their peers for acceptance [18]. Consequently, children who are depressed may resort to bullying and aggressive behaviors, as studies have shown [21,22,23]. Aggressive behavior and bullying are commonly reported by youth with psychological problems, including depressive symptoms [24].

Academic disengagement represents another potential mediator that amplifies the association between experiences in racial discrimination and bullying behavior in adolescents. Not surprisingly, perceived discrimination influences academic outcomes, and youth who are racially discriminated against have the tendency of feeling disengaged academically at their school [25,26]. As reported in Leath et al.’s [26] study, which was composed of adolescent samples of Black working-class and White middle-class backgrounds, racial discrimination was negatively related to academic curiosity and persistence. In turn, students who are academically disengaged might be inclined to turn to risky and delinquent behaviors, including bullying perpetration. To illustrate, findings from both Demosthenous et al.’s [27] and Hemphill et al.’s [28] studies suggest that academic decline was positively correlated with problem behaviors, including bullying perpetration.

### 1.2. The Present Study

Although a limited number of empirical studies suggest that adolescents who experience racial discrimination may be at an elevated risk of engaging in aggressive behaviors [6], research to date has not investigated mediators, for example, depression and academic disengagement, which might amplify the association between racial discrimination and bullying. It is rare that adolescents who are discriminated against exhibit aggressive behavior immediately; rather, they are likely to experience psychosocial problems, such as depression and academic disengagement, which might potentially increase their risk of bullying behavior. The present study aims to examine the relationship between racial discrimination and bullying perpetration. We hypothesize that experiences in racial discrimination would be directly and positively associated with bullying perpetration. The study also proposes and empirically tests pathways by which racial discrimination might be associated with bullying perpetration among Black and White adolescents. We hypothesize that racial discrimination would be related to higher bullying perpetration via greater depression and academic disengagement. For this study, the measures for the variables, including the child’s bullying, were derived exclusively from parental reports. Parents are considered important sources of information since they tend to observe the child over long periods and in various contexts [29]. Additionally, researchers have long supported parental involvement in school bullying prevention efforts [30]. Thus, it might also be important to understand whether parents are a valid source of information not only for researchers but also for teachers or practitioners who normally discuss bullying with the adolescents’ parents. Since parents are an important part of the child and adolescent development, it is important to understand whether there are similar relationships among the study variables as having been found in youth self-reports.

## 2. Materials and Methods

### 2.1. Sample and Procedure

The National Survey of Children’s Health (NSCH) is a national survey that provides data on multiple, intersecting aspects of child physical and mental health and their family context. In 2003, 2007, and 2011–2012, the National Center for Health Statistics at the Centers for Disease Control and Prevention conducted a telephone survey under the sponsorship and direction of the Maternal and Child Health Bureau. In 2016, the U.S. Census Bureau administered the telephone survey using web- and paper-based instruments and consolidated content from the two surveys. The present study uses the 2016 NSCH dataset.

NSCH aimed to produce national and state-level data on the physical and emotional health of U.S. children (ages 0–17 years). Data were collected on family interactions, parental health, and school/after-school experiences. Households were contacted by mail based on random selection to identify those with one or more children, 17 years of age or younger, and in each household, one child was randomly selected to be the participant in the survey. Data collection occurred between June 2016 to February 2017. A letter was sent to randomly selected households, which contained an invitation to participate in the survey through the web; non-responders received the mailing, which consisted of a paper instrument to complete and return via postal mail. A total of 50,212 surveys were completed nationally. The overall weighted response rate was 40.7%. Nine hundred and eighty-five surveys were collected per state, and the results of the survey were weighted to represent the population of non-institutionalized children who live in housing units nationally and in each state. The total study sample was 16,126 parents, consisting of Black (*n* = 1213) and White (*n* = 14,913) parents of adolescents, aged 12 to 17. The socio-demographic characteristics of the adolescent and the participants are presented in Table 1. Among the 16,126 participants, 51% were male and 49% were female, and the mean age was 14.72 years old (*SD* = 1.69, range 12–17). The mean age of the caregiver for the total sample was 47.52 years old (*SD* = 7.47, range 20–75), and the majority of caregivers (67.6%) were female.

### 2.2. Measures

*Bullying perpetration* was measured with one item reported by the caregivers, “This child bullies others, picks on them, or excludes them.” Response options were *not at all true (1)*, *somewhat true (2)*, and *definitely true (3)*.

*Racial discrimination* was measured with one item, “Has this child ever experienced being treated or judged unfairly because of his or her race or ethnic group” Response options were *no (0)* and *yes (1)*.

*Depression* was measured with one item, which asks the caregiver about the child’s health: “Has a doctor or other health care provider ever told you that your child has depression?” Response options were *no (0)* and *yes (1)*.

*Academic disengagement* was measured with three items, “In general, how would you describe this child (a) that this child shows interest and curiosity in learning new things, (b) that this child works to finish tasks he or she starts, and (c) that this child cares about doing well in school.” Response options were *definitely true (1)*, *somewhat true (2)*, and *not at all true (3)*. A composite score was calculated, with higher scores indicating a lower level of academic engagement. Cronbach’s alpha was 0.76.

Socio-demographic characteristics of caregiver and child participants were controlled for in the model. Covariates for the study included the *caregiver’s* and *child’s age* and *sex. Caregiver* and *child’s ages* were measured as a continuous variable. *Caregiver and child’s sexes* were coded dichotomously (*male* [1] and *female* [2]).

### 2.3. Data Analyses

We first estimated descriptive analyses to calculate variable distributions in multivariate analyses and conducted bivariate correlations between all the potential variables. Next, we employed structural path analyses with Mplus 7.0 [31] to test the hypothesized path model, after controlling for the covariates.

To assess model fit, we used multiple indices, including Root Mean Squared Error of Approximation (RMSEA), Standardized Root Square Mean Residual (SRMR), Comparative Fit Index (CFI), and Tucker–Lewis Index (TLI). Although the chi-square test has been used as one of many other indices of model fit, we should note that the chi-square values are highly sensitive to sample size and other biases [32]. Therefore, a significant chi-square is not, by itself, a reason to modify a model, if other indices can provide a good fit [33]. This study relied on a standard cutoff recommendation for RMSEA, SRMR, CFI, and TLI [34]. For RMSEA and the SRMR, values less than 0.05 indicated a good fit. For TLI and CFI, values greater than or equal to 0.90 indicated an acceptable model fit.

The percentages of missing data at the variable level were less than 2%. The MCAR test is only available for continuous variables, and the study variables are mostly categorical. Academic disengagement is a continuous variable, but the MCAR test for it did not reject the null hypothesis. The missing values in our study are relatively small (less than 2%). Graham [35] argued that if missing values are less than 5%, biases and low power are both likely to be inconsequential. Robust MLR estimation was used for biases due to missing values. Only *bullying perpetration* is the variable with non-normal distribution (skewness = 5.07, kurtosis = 27.33). Therefore, the robust maximum likelihood estimator (MLR) was used because it does not require the assumption of normality and provides mean- and variance-adjusted chi-square test statistics and corrected standard errors [31]. Tests of indirect effects based on Mplus estimation assessed the strength of the mediated relationships [31].

## 3. Results

### 3.1. Descriptive Statistics

Approximately 2.6% of the total participants reported that their child experienced racial discrimination. Among the total sample, only 1.5% of White adolescents reported experiencing racial discrimination, while 16.3% of Black adolescents reported experiencing racial discrimination. Of the total, 9.2% of adolescents have ever been told by a doctor or a health care provider they have depression, and White adolescents (9.2%) reported more depression than Black adolescents (8.5%). The mean for the mother’s report of child’s bullying was 1.06 (*SD* = 0.27, range 1–3) and for academic disengagement was 3.81 (*SD* = 1.27, range 1–9).

### 3.2. Correlation Analysis

Correlation analysis results are displayed in Table 2. Racial discrimination (*r* = 0.042, *p* < 0.001), depression (*r* = 0.126, *p* < 0.001), and academic disengagement (*r* = 0.192, *p* < 0.001) were all associated with an increase in bullying. We also calculated the tolerance and VIF for each independent variable in the model. All variables indicated a value of less than 10, implying the little possibility of multicollinearity.

### 3.3. Path Analysis

Path analysis was employed for White and Black adolescents to test the hypothesized relationships. The hypothesized models for both races were adjusted for the covariates and were tested. The model for both races fit the data well: for White people, χ^2^ (6) = 57.152, *p* = 0.000, CFI = 0.983, TLI = 0.950, RMSEA = 0.024 (90% CI = 0.019 to 0.030), and SRMR= 0.011; for Black people, χ^2^ (6) = 12.66, *p* = 0.048, CFI = 0.985, TLI = 0.954, RMSEA = 0.031 (90% CI = 0.002 to 0.056), and SRMR= 0.015. Figure 1 presents the standardized path estimates for White people and Figure 2 presents the standardized path estimates for Black people.

In the White adolescents model, after adjusting for the covariates, racial discrimination was positively associated with bullying perpetration (β = 0.023, *p* = 0.035). Adolescents’ depression (β = 0.085, *p* = 0.000) and academic disengagement (β = 0.188, *p* = 0.000) were also positively associated with bullying perpetration.

In the Black adolescents model, after adjusting for the covariates, racial discrimination was positively associated with bullying perpetration (β = 0.064, *p* = 0.029). Adolescents’ depression (β = 0.100, *p* = 0.001) and academic disengagement (β = 0.178, *p* = 0.000) were also positively associated with bullying perpetration.

Three indirect paths for the White adolescents model were significant as follows: (a) racial discrimination → depression → bullying perpetration (indirect β = 0.002, *p* = 0.015; 95% CI = 0.001 to 0.004), (b) racial discrimination → academic disengagement → bullying perpetration (indirect β = 0.005, *p* = 0.005; 95% CI = 0.002 to 0.008), and (c) racial discrimination → depression → academic disengagement → bullying perpetration (indirect β = 0.001, *p* = 0.010; 95% CI = 0.001 to 0.002). The indirect paths for the Black adolescents model were significant as follows: racial discrimination → depression → academic disengagement → bullying perpetration (indirect β = 0.004, *p* = 0.002; 95% CI = 0.001 to 0.008), (b) racial discrimination → depression → bullying perpetration (indirect β = 0.008, *p* = 0.004; 95% CI = 0.001 to 0.016).

Regarding the covariates, only the child’s age was negatively related to bullying perpetration for both races (for Whites: β = −0.034, *p* = 0.000; for Blacks: β = −0.078, *p* = 0.032), indicating that younger adolescents were more likely to be involved in bullying perpetration.

## 4. Discussion

The present study aimed to provide a more comprehensive understanding of the relationship between racial discrimination and bullying behavior, and whether depression and academic disengagement mediated this association among White and Black adolescents. For adolescents in both racial groups, racial discrimination appears to be a chronic stressor, which has a positive association with the psychological state (i.e., depression). This finding was consistent with prior study findings [18,19] and the proposed study hypothesis. Irrespective of racial or ethnic identity, adolescents who are repeatedly mistreated by their peers in school because of their racial identity may be prone to displaying psychological distress and show signs of depressive symptoms. Further, depression was positively associated with bullying perpetration in our study, which was consistent with our proposed hypothesis and the research literature [21,22,23]. Overall, this finding seems to indicate that depression could amplify how racial discrimination might be positively associated with bullying perpetration for both White and Black adolescents.

To our surprise, however, the link between racial discrimination and academic disengagement was significant for White adolescents but not for Black adolescents. This was unexpected as we had hypothesized that racial discrimination would contribute to lower academic engagement for both White and Black adolescents. Notwithstanding such unexpected results, a possible explanation might be that racial discrimination against Black people tends to be more frequent and widespread, historically embedded, and taken for granted in the society, that part of it is almost “invisible” and under the radar when caregivers are asked to report on it. Because Black adolescents are at a significantly higher risk of racial discrimination, they may not be as impacted by it academically as White adolescents are as they learn early on that they will experience racial discrimination. However, the psychological detriments of racial discrimination have been documented in the research literature [36], which might explain why Black adolescents show signs of depression when they are victims of racial discrimination. Another possible explanation might be, as our data suggest, that the link is indirect via depression for Black adolescents. Black students who are negatively affected mentally (e.g., showing signs of depressive symptoms) by being a target of racial discrimination might show a decrease in academic engagement in school. Overall, our findings suggest that specific attention to stressors such as racial discrimination in research on bullying is warranted.

### 4.1. Limitations and Implications for Research

The present study has some notable strengths. The large, nationally representative study sample is useful in statistical analyses and would allow for generalizing the findings. In addition, as previously mentioned, parents’ and caregivers’ observations can shed light on children’s and adolescents’ behaviors. Despite these strengths, some limitations of the study need to be acknowledged. The cross-sectional study design does not allow us to draw a conclusion on the temporal ordering of the study’s main variables, namely discrimination, depression, academic disengagement, and bullying perpetration. Follow-up research should try to replicate the present findings by using longitudinal data with at least three measurement points to further substantiate the mediating relationships tested in the present study. In addition, racial discrimination, depression, and bullying perpetration were measured by one single item each. The use of single-item measures is often accompanied by typical measurement problems (e.g., low content validity, sensitivity, and lack of a measure of internal consistency reliability). Future research that would help overcome such problems should use validated scales to investigate those relationships. Moreover, given that children’s bullying occurs mostly outside of the home, parental reporting of a child’s bullying can be underestimated. However, research using both self-report surveys and teacher reports suggests that although teachers can provide some important information, parent observations are more useful for screening for some types of problems, such as behavior problems and conduct problems [29,37]. Further, the present study included only parent reports, which, despite its strength, is also a weakness in that the validity of the findings is questionable. Additionally, racial discrimination was not significantly associated with academic disengagement among Black adolescents, which might be due to the small effect size. Finally, the effects found in the study were small because parent reports were used instead of self-reports. Future research should combine different information sources (e.g., peer reports, teacher reports, children’s self-reports) to determine the accuracy of parent reports of the investigated relationships in the present study.

### 4.2. Implications for Practice

The above limitations aside, the present findings have some practical implications. Our findings suggest that racial discrimination needs to be addressed in bullying prevention as well as in student mental health promotion. The establishment and maintenance of a safe and healthy school should include efforts in reducing racial discrimination, considering its links to depression and bullying perpetration among students. In addition to its direct negative effect on mental health [38], racial discrimination might also have a further negative impact on students’ mental health by increasing bullying perpetration in school, as our findings suggest, which, in turn, should result in more bullying victimization, and both bullying perpetration and victimization have been found to predict mental health problems according to a growing body of longitudinal studies [39,40].

Fahd and Venkatraman [41] propose an education system framework to address racial discrimination in school. The macro–national level includes changes in the legal and public policies. The meso–organizational level involves management strategies at the school level, such as anti-racism policy of schools, educational sessions for school staff to prevent and counter racism and attain cross-cultural knowledge, designing curriculum content, rigorous playground monitoring, and involving parents to counter racism. The micro–individual level includes educating students to be aware of their rights regarding racial discrimination, bystander training, assertiveness training, restorative practices, and encouraging students to engage in tackling racial problems. Efforts in achieving racial/ethnic inclusion, equity, and fairness in school are not only essential in promoting social justice and counteracting racism and ethnic segregation, but the present findings propose that these interventions are also important to promote mental health and prevent bullying in school.

## 5. Conclusions

Research on race and ethnicity concerning children’s bullying has increased over the years [42], and findings from the present study underscore a positive association between experiencing racial discrimination and engaging in bullying among adolescents, which suggests that efforts to prevent bullying need to consider how experiences in racism and racial discrimination might affect students’ behavior and relations with others in the school. Although beyond the scope of the present study, it is possible that for some students, especially racial and ethnic minorities, bullying others or acting out aggressively is perceived to be a way of dealing with racial discrimination. That being said, bullying behavior should never be justified or condoned; however, it is important that racial discrimination is being considered in bullying prevention efforts, as racism and bullying seem to go hand in hand.

## Figures and Tables

**Figure 1 ijerph-19-07084-f001:**
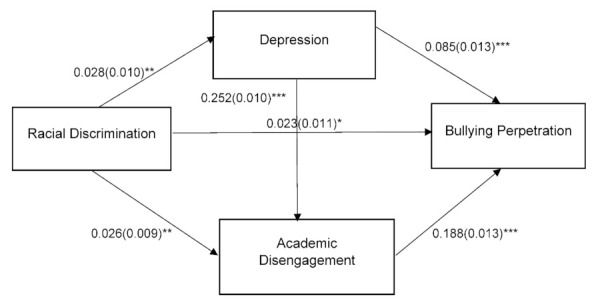
Pathways from Racial Discrimination to Bullying Perpetration for White Adolescents. Note: Controlled for child sex, child age, caregiver’s sex, and caregiver’s age; standardized coefficients are presented, and standard error is in the parentheses. * *p* < 0.05; ** *p* < 0.01; *** *p* < 0.001.

**Figure 2 ijerph-19-07084-f002:**
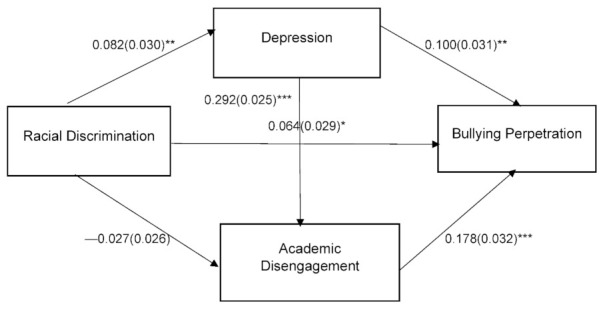
Pathways from Racial Discrimination to Bullying Perpetration for Black Adolescents. Note: Controlled for child sex, child age, caregiver’s sex, and caregiver’s age; standardized coefficients are presented, and standard error is in the parentheses. * *p* < 0.05; ** *p* < 0.01; *** *p* < 0.001.

**Table 1 ijerph-19-07084-t001:** Descriptive Statistics of the Total Sample.

Variables	Total (*n* = 16,126)	White (*n* = 14,913)			Black (*n* = 1213)	
	*n* (%)	*M* (*SD*)	*n* (%)	*M* (*SD*)	*n* (%)	*M* (*SD*)
Bullying perpetration (1–3)		1.06 (0.27)		1.06 (0.25)		1.08 (0.31)
Racial discrimination	409 (2.6)		221 (1.5)		188 (16.3)	
Depression	1477 (9.2)		1374 (9.2)		103 (8.5)	
Academic disengagement (1–9)		3.81 (1.27)		3.80 (1.26)		3.94 (1.36)
Child age (10–17)		14.72 (1.69)		14.73 (1.69)		14.67 (1.66)
Child sex						
Male	8232 (51.0)		7600 (51.0)		632 (52.1)	
Female	7894 (49.0)		7313 (49.0)		581 (47.9)	
Caregiver’s age (20–75)		47.52 (7.47)		47.51 (7.26)		47.61 (9.71)
Caregiver’s sex						
Male	5127 (32.4)		4856 (33.1)		271 (23.2)	
Female	10,697 (67.6)		9799 (66.9)		898 (76.8)	

**Table 2 ijerph-19-07084-t002:** Bivariate Correlations of the Main Study Variables.

	1	2	3	4	5	6	7	8	9
1. Bullying perpetration	-								
2. Racial discrimination	0.042 ***	-							
3. Depression	0.126 ***	0.032 ***	-						
4. Academic disengagement	0.192 ***	−0.032 ***	−0.262 ***	-					
5. Child age	−0.026 **	0.001	−0.092 ***	0.003	-				
6. Child sex	−0.020 *	0.008	−0.058 ***	−0.164 ***	0.003	-			
7. Caregiver’s age	−0.022 ***	−0.001	−0.013	−0.016 *	0.212 ***	−0.004	-		
8. Caregiver’s sex	0.020 *	−0.016 *	−0.059 ***	0.042 ***	0.006	0.005	−0.209 ***	-	
9. Race (Black/White)	−0.023 **	0.242 ***	−0.006	−0.030 ***	0.009	0.006	−0.003	−0.056 ***	-

* *p* < 0.05; ** *p* < 0.01; *** *p* < 0.001.

## Data Availability

Not applicable.

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
