# Peer review of "Racial Discrimination to Bullying Behavior among White and Black Adolescents in the USA: From Parents’ Perspectives"

_ijerph, 2022, doi:10.3390/ijerph19127084_

Round 1

Reviewer 1 Report

So, this can't be published in its current form because it has a place holder for the discussion section instead of an actual discussion. The conclusions do include much of what I'd put in a discussion so this is really an editing issue more than anything else. However, with that correction it is an interesting an important piece and shoudl be published. There are a few things I'd like the authors to think about:

I think that one of the limitations that the authors might mention and perhaps speculate about is that the data presented here are old. The most recent dates on the surveys used are 2012. That feels like a lifetime ago. Importantly, it was arguably a better, kinder moment in US history. Surveys from that era show that people believed race relations were improving and there was certainly widespread concern about bullying. Politics, culture wars, Covid, and many other factors may well have led to increases in discrimination and bullying. Both physical and emotional attacks on people have skyrocketed. The FBI reported about double the hate crimes in 2020 as it did in 2012. I think this probably increases the importance of this work but it needs a comment.

The authors discuss this but I got to say I too am skeptical of the relationship between parents' reports of bullying and discrimination and kids actual experiences of them. I think that parent reports are likely to greatly underestimate the incidence of both. 

The fact that academic disengagement does not significantly link with discrimination for Black students is interesting. It would be nice if the authors included something about the size of this effect in the paragraph where it is discussed. 

Author Response

So, this can't be published in its current form because it has a place holder for the discussion section instead of an actual discussion. The conclusions do include much of what I'd put in a discussion so this is really an editing issue more than anything else. However, with that correction it is an interesting an important piece and should be published. There are a few things I'd like the authors to think about:

RESPONSE: I deeply apologize for this. This was a typo and the “Conclusion” section should have been in the Discussion section. Corrections were made, and a separate Conclusion section was added. Thank you very much for pointing out this embarrassing mistake

I think that one of the limitations that the authors might mention and perhaps speculate about is that the data presented here are old. The most recent dates on the surveys used are 2012. That feels like a lifetime ago. Importantly, it was arguably a better, kinder moment in US history. Surveys from that era show that people believed race relations were improving and there was certainly widespread concern about bullying. Politics, culture wars, Covid, and many other factors may well have led to increases in discrimination and bullying. Both physical and emotional attacks on people have skyrocketed. The FBI reported about double the hate crimes in 2020 as it did in 2012. I think this probably increases the importance of this work but it needs a comment.

RESPONSE: The data used for this study was the 2016 NSCH dataset. We apologize for not being clear about it. In the abstract, it states, “Data were derived from the 2016 National Survey of Children’s Health….” In the Methods section, we included the following statement, “The present study uses the 2016 NSCH dataset.”

The authors discuss this but I got to say I too am skeptical of the relationship between parents' reports of bullying and discrimination and kids actual experiences of them. I think that parent reports are likely to greatly underestimate the incidence of both.

RESPONSE: We completely understand this concern, and the usage of parental report to measure bullying perpetration was included in the limitations section. However, there are several studies that have utilized this dataset to explore children’s bullying.  

The fact that academic disengagement does not significantly link with discrimination for Black students is interesting. It would be nice if the authors included something about the size of this effect in the paragraph where it is discussed. 

RESPONSE: In the “4.1. Limitations and Implications for Research” section, we included the statement, “Additionally, racial discrimination was not significantly associated with academic disengagement among Black children, which might be due to the small effect size.”

Reviewer 2 Report

Dear Authors,

Congratulations of your paper.

Although the interest of the subject, some improvements are needed, namely:

-  Authors should highlight in the Introduction section the main aims of the study.

- Hypothesis should be supported by the literature review.

- What type of sampling did authors use? Please explain it.

- A discussion section is missing. Authors should insert it. As they refer in their text "Authors should discuss the results and how they can be interpreted from the perspective of previous studies and of the working hypotheses. The findings and their implications should be discussed in the broadest context possible. Future research directions may also be highlighted." This needs to be inserted in your manuscript.

- Authors should highlight, clearly, the main theoretical contribution of this study to this research field

Author Response

Authors should highlight in the Introduction section the main aims of the study.

RESPONSE: We added the following statement, “The present study addresses the following research hypotheses: (a) racial discrimination would be significantly and positively associated with bullying perpetration among White and Black adolescents, as reported by the parents, and (b) racial discrimination would be positively related to depression, which would be positively associated with academic disengagement and bullying perpetration for both White and Black adolescents, as reported by their parents.” This is located at the end of “1.2. The Present Study” section.

Hypothesis should be supported by the literature review.

RESPONSE: The hypotheses we proposed are aligned with the literature review.

What type of sampling did authors use? Please explain it.

RESPONSE: This study utilized a secondary dataset, the National Survey of Children’s Health (NSCH). The authors did not collect the data. The second paragraph of “2.1. Sample and Procedure” describes how data were collected in NSCH. We did include the following: “Data collection occurred between June 2016 to February 2017. A letter was sent to randomly selected households, which contained an invitation to participate in the survey through the web; non-responders received the mailing, which consisted of a paper instrument to complete and return via postal mail.”

A discussion section is missing. Authors should insert it. As they refer in their text "Authors should discuss the results and how they can be interpreted from the perspective of previous studies and of the working hypotheses. The findings and their implications should be discussed in the broadest context possible. Future research directions may also be highlighted." This needs to be inserted in your manuscript.

RESPONSE: I deeply apologize for this. This was a typo and the “Conclusion” section should have been in the Discussion section. Corrections were made, and a separate Conclusion section was added. Thank you very much for pointing out this embarrassing mistake.

Authors should highlight, clearly, the main theoretical contribution of this study to this research field.

RESPONSE: We included the following statement in “1.2. The Present Study” section: “Although a limited number of empirical studies suggest that adolescents who experience racial discrimination may be at an elevated risk of engaging in aggressive behaviors [6], research to date has not investigated mediators, for example, depression and academic disengagement, which might amplify the association between racial discrimination and bullying. It is rare that adolescents who are discriminated against exhibit aggressive behavior immediately; rather, they are likely to experience psychosocial problems, such as depression and academic disengagement, which might potentially increase their risk of bullying behavior.”

Reviewer 3 Report

See attached file, please.

Author Response

As for the participants in this study everything is confusing and ambiguous. Why? The authors state that parents are the agents of information regarding the variables analyzed in this study. However, later, in the Participants section, the authors claim to recruit a sample of children, so what are we talking about?

RESPONSE: I apologize for the confusion. In the “2.1. Sample and Procedure” section, we changed “child participants to “children of the participants”.

In any case, since the beginning of this ms. (Abstract) authors should clarify and describe:

Type of sampling conducted to recruit parents. Indicate percentage of men or women, age range, mean and SD for the two groups examined: Black and White. Type of sampling conducted to recruit the children who participated in this study. Indicate percentage of girls or girls, age range, mean and SD for the two groups examined: Black and White.

RESPONSE: The following was included in the Abstract: “…between June 2016 to February 2017. A letter was sent to randomly selected households, who were invited to participate in the survey. The caregivers consisted of 66.9% females and 33.1% males for the White sample whose mean age was 47.51 (SD = 7.26), and 76.8% females and 23.2% males for the Black sample whose mean age was 47.61 (SD = 9.71). In terms of the children, 49.0% were females among the White sample whose mean age was 14.73 (SD = 1.69). For Black children, 47.9% were females and the mean age was 14.67( SD = 1.66).”

The Black and White parent sample is totally unbalanced. This is surprising and a robust explanation is necessary.

RESPONSE: The racial distribution of the sample was unbalanced, as this is representative of the population in the USA, which would be 50.2% White (non-Hispanic) and 13.3% Black, non-Hispanic). For more information, please refer to https://nschdata.org/browse/survey/results?q=8386

Table 1 is confusing, ambiguous.

RESPONSE: We believe because of spacing, etc. it was confusing. We fixed the table so that it would be easier to interpret.

The authors do not provide robust reasons to justify this study; what is the novelty, improvement or scientific advance of this ms.? I do not know.

RESPONSE: We included the following statement in “1.2. The Present Study” section: “Although a limited number of empirical studies suggest that adolescents who experience racial discrimination may be at an elevated risk of engaging in aggressive behaviors [6], research to date has not investigated mediators, for example, depression and academic disengagement, which might amplify the association between racial discrimination and bullying. It is rare that adolescents who are discriminated against exhibit aggressive behavior immediately; rather, they are likely to experience psychosocial problems, such as depression and academic disengagement, which might potentially increase their risk of bullying behavior.”

The authors should end the introduction section indicating what the main objective of this study is, as well as what the specific objectives are. In addition, the authors must formulate a hypothesis for each specific objective, which must be robustly accepted or rejected in the discussion section.

RESPONSE: We included the following statements in the 1.2. The Present Study” section: “The present study aims to examine the relationship between racial discrimination and bullying perpetration. We hypothesize that experiences in racial discrimination would be directly and positively associated with bullying perpetration. The study also proposes and empirically tests pathways by which racial discrimination might be associated with bullying perpetration among Black and White children. We hypothesize that racial discrimination would be related to higher bullying perpetration via greater depression and academic disengagement.”

The correlation coefficients presented in Table 2 must be described (Results) and correctly interpreted (Discussion) according to the magnitude or size of the effect of these coefficients. In this sense, most correlation coefficients are insignificant, they have no empirical relevance because they are less than or equal to .20 (small effect size; Cohen, 1988). So?

RESPONSE:  All correlations between study variables in the present study were significant except for control variables, and we described this result in the text. In addition, Pearson’s correlation (=r) refers to the degree to which a pair of variables is linearly related. However, the Cohen’s effect size (=d) quantifies some difference between two groups (e.g. the difference between the means and standard deviations of two datasets). Therefore, these two are related, but not directly related or substituted. Correlation analysis in this research is to identify the linear relationship, significant connections, and possibility of multicollinearity between study variables.

Round 2

Reviewer 2 Report

Dear Authors,

Congratulations.

Improvements were made accordingly.